# Elucidating the Role of *KRAS*, *NRAS*, and *BRAF* Mutations and Microsatellite Instability in Colorectal Cancer via Next-Generation Sequencing

**DOI:** 10.3390/cancers17132071

**Published:** 2025-06-20

**Authors:** Marta Rada Rodríguez, Bárbara Angulo Biedma, Irene Rodríguez Pérez, Javier Azúa Romeo

**Affiliations:** 1Department of Human Anatomy and Histology, Universidad de Zaragoza, Calle Pedro Cerbuna, 50009 Zaragoza, Spain; radarodriguezmarta@gmail.com; 2Area of Molecular Biology, Analiza, 28008 Madrid, Spain; 3Area of Pathology, Analiza, 28008 Madrid, Spain; irodriguez@analizalab.com; 4Chief of Pathology, Analiza, 28008 Madrid, Spain

**Keywords:** *BRAF*, biomarker, genetic mutations, DNA mismatch repair, *KRAS*, microsatellite instability, molecular profiling, next-generation sequencing, *NRAS*, precision oncology, sporadic colorectal cancer

## Abstract

Background: Next-generation sequencing (NGS) in sporadic colorectal adenocarcinoma (CRC) represents an ambitious diagnostic approach that may pose considerable challenges. There are limitations regarding the indication and availability of this technique, which may indirectly influence the outcome of the underlying disease. Objective: To identify and analyze the frequency of mutations in commonly altered genes in this pathology, assess their correlation, and highlight the need for a personalized therapeutic approach based on the molecular profile of the tumor. Additionally, we evaluated the importance of early genomic analysis using NGS in comparison to other techniques (real-time PCR and immunohistochemistry), aiming to optimize diagnosis in terms of turnaround time, tissue preservation, and cost-effectiveness. Our findings reinforce the role of *KRAS* as the most prevalent mutation, typically associated with microsatellite-stable tumors, whereas a mutated *BRAF* gene is more frequently associated with MSI.

## 1. Introduction

Colorectal cancer (CRC) is one of the most common malignancies and the leading cause of cancer-related death in the digestive system. Most CRCs arise through the chromosomal instability (CIN) pathway, typically initiated by inactivation of the APC gene, characterized by a loss of tumor suppressor genes, activation of oncogenes, and widespread genetic alterations. A less frequent pathway (10–15%), referred to as the mutator phenotype or microsatellite instability (MSI) pathway, results from mutations in genes encoding DNA mismatch repair (MMR) proteins. This leads to the accumulation of numerous genetic errors. The majority of hereditary nonpolyposis colorectal cancers (HNPCCs) and a proportion of sporadic CRCs (10–15%) exhibit MSI, a molecular hallmark useful for detection [1,2].

CRC tumors harboring *BRAF* mutations and/or high microsatellite instability (MSI-H) represent clinically and pathologically distinct subgroups with different prognoses [3,4]. Approximately 2–15% of CRCs exhibit *BRAF* mutations, with the V600E mutation (c.1799T > A, p.Val600Glu) being the most prevalent [5,6]. MSI-H is observed in about 12–15% of all CRCs and around 4% of metastatic CRCs. MSI-H status alone has generally been associated with a more favorable prognosis, as these tumors are typically diagnosed at earlier stages and are more likely to respond to immunotherapy, particularly to immune checkpoint inhibitors targeting the PD-1/PD-L1 axis [7].

Furthermore, the presence of the *BRAF* V600E mutation has been associated with tumors located predominantly in the right colon, showing poor differentiation and a higher prevalence among female patients [6]. This mutation has also been specifically correlated with methylation of the MLH1 gene promoter, a common mechanism in sporadic MSI-H tumors [8,9].

*BRAF* and MSI status has been significantly linked to metastatic dissemination patterns and prognosis in CRC patients. *BRAF* mutations, particularly in microsatellite-stable (MSS) tumors, have been associated with worse outcomes, including decreased overall survival, disease-specific survival, and progression-free survival [10]. An accurate assessment of *BRAF* and MSI status is therefore crucial not only for prognostic stratification but also for guiding therapeutic decision-making in CRC [6]. While the presence of *BRAF* mutations predicts a limited response to certain targeted therapies—such as cetuximab and panitumumab—efficacy can be improved when combined with anti-EGFR monoclonal antibodies. A combination of encorafenib, cetuximab, and binimetinib has shown significantly prolonged overall survival and higher response rates compared to standard therapy in patients with metastatic CRC harboring *BRAF* V600 mutations [11].

In particular, the detection of high-frequency microsatellite instability (MSI-H), associated with defects in DNA mismatch repair (dMMR), has become an essential biomarker for determining eligibility for immune checkpoint inhibitor (ICI) therapies, which have demonstrated efficacy in neoplasms with a high mutational burden. NGS-based methods, especially comprehensive genomic profiling (CGP) assays, offer the advantage of simultaneously analyzing MSI status alongside other genomic alterations in a single test, including through liquid biopsies, thereby enabling a less invasive and more comprehensive approach [12].

Likewise, MSI-H status has emerged as a key factor indicating reduced benefits from conventional chemotherapy, particularly in stage II and III tumors, which show a limited response to adjuvant 5-FU-based regimens [13].

MSI status can be determined using polymerase chain reaction (PCR) or immunohistochemistry (IHC) to assess MMR protein expression. The detection of *BRAF* mutations—typically the V600E variant—can be carried out using real-time PCR-based kits and, to a lesser extent, by Sanger sequencing or IHC with specific antibodies. Next-generation sequencing (NGS) has been proposed as a comprehensive tool for genomic profiling, enabling the simultaneous detection of multiple gene mutations, including *BRAF*, as well as an assessment of MSI status and tumor mutational burden (TMB) [14].

Despite increasing knowledge of the roles of *BRAF* and MSI in CRC, challenges remain in interpreting results and integrating this information into optimal clinical practice, particularly regarding metastatic patterns and therapeutic responsiveness [15].

NGS has proven to be a valuable tool for the molecular diagnosis and treatment planning of CRC, especially in patients with MSI or deficient mismatch repair (dMMR) protein expression. Its advantages include providing a comprehensive overview of the tumor’s molecular profile, identifying patients unlikely to benefit from chemotherapy, selecting those who are likely to respond to immunotherapy, enabling earlier access to targeted therapies (currently approved for advanced or metastatic stages), and enhancing overall prognostic accuracy.

Unlike lung cancer, colorectal cancer (CRC) still has limited precision treatment options. However, the receptor ERBB2 (also known as HER2) has emerged as a promising therapeutic target in CRC. Oncogenic alterations in HER2, as well as in its dimerization partner HER3, may directly influence sensitivity to HER2-targeted therapies. This underscores the importance of including these markers in the molecular profiling of CRC to identify patients who may benefit from specific targeted treatments [16,17].

By facilitating more personalized and effective treatment from the early stages, the upfront use of NGS has the potential to improve overall outcomes in CRC patients [18,19].

## 2. Materials and Methods

### 2.1. Study Design

An observational, retrospective, and cross-sectional study was conducted on patients diagnosed with sporadic colorectal cancer (sCRC). Mutations in the *KRAS*, *NRAS*, and *BRAF* genes, as well as microsatellite instability (MSI), were analyzed using next-generation sequencing (NGS). Additionally, the NGS results were compared with conventional techniques, such as real-time PCR and immunohistochemistry, using pseudoanonymized cases diagnosed in Spain and provided by the pathology laboratory Analiza.

### 2.2. Population and Case Selection

A total of 648 cases with histologically confirmed colon adenocarcinoma were retrospectively and cross-sectionally reviewed. Among them, 166 cases had partial molecular data available, and 42 cases were ultimately included in the final analysis, as they contained all molecular markers relevant to this study.

Case selection was conducted in two phases. Initially, patients diagnosed between 1 January 2024 and 31 May 2024 were screened using broader inclusion criteria. Subsequently, the retrospective search was extended up to 5 February 2025, including cases diagnosed from 1 January 2022, onwards, applying stricter criteria to obtain a more homogeneous cohort with complete molecular profiles.

#### 2.2.1. Inclusion Criteria

Histopathologically confirmed diagnosis of sporadic colorectal cancer.Availability of complete sequencing data for *KRAS*, *NRAS*, *BRAF*, and MSI.Sufficient tumor tissue for molecular analysis.

#### 2.2.2. Exclusion Criteria

Cases of hereditary non-polyposis colorectal cancer (HNPCC) or Lynch syndrome at the time of diagnosis.Patients lacking complete information regarding the genes studied or MSI status.Inadequate or degraded samples unsuitable for molecular testing.

### 2.3. Molecular Analysis Procedures

#### 2.3.1. DNA Extraction

Tumor DNA was extracted from formalin-fixed, paraffin-embedded (FFPE) tissue samples using standardized commercial kits (QIAamp DSP FFPE Tissue Kit, Qiagen, Hilden, Germany). DNA concentration and purity were assessed by fluorometric quantification.

#### 2.3.2. Next-Generation Sequencing (NGS) with Action OncoKitDx^®^ Panel

The Action OncoKitDx panel (Health in Code Group, Valencia, Spain) is designed to analyze genetic alterations across 59 genes relevant to solid tumor development. The panel detects point mutations (substitutions, deletions, insertions), copy number variations, and rearrangements, all of which have diagnostic, prognostic, and therapeutic significance, including potential as druggable targets or predictive biomarkers for approved or investigational targeted therapies. It also includes MSI analysis, relevant in the context of immunotherapy, and pharmacogenetic profiling of variants associated with chemotherapy toxicity or efficacy.

The Action OncoKitDx panel allows full exon sequencing of *KRAS*, *NRAS*, and *BRAF* genes. MSI is assessed through a panel of 110 microsatellite regions. A minimum of 99 evaluable markers is required, and classification is based on the percentage of unstable markers as follows: high MSI (31–100%, MSI-H), low MSI (21–30%, MSI-L), microsatellite stable (0–17%, MSS), and inconclusive results (18–20%)The panel is prepared using an automated workflow with the Magnis Dx NGS Prep System (Agilent, Santa Clara, CA, USA). After DNA extraction, samples undergo enzymatic fragmentation and enrichment of target regions via hybridization with capture probes using SureSelectXT HS technology, according to the manufacturer’s instructions. High-throughput NGS is performed on the NextSeq 550 platform (Illumina, San Diego, CA, USA) using paired-end sequencing (2 × 75 bp) and cyclic reversible termination chemistry. This allows for the detection of point mutations and larger sequence alterations in the targeted genes.Bioinformatic analysis is performed using a dedicated pipeline through the Data Genomics platform “www.datagenomics.es (accessed on 11 June 2025)”. It includes alignment of reads to the reference genome (GRCh37/hg19), quality filtering, and variant calling. Variant nomenclature follows the guidelines of the Human Genome Variation Society “HGVS; www.hgvs.org (accessed on 11 June 2025)”.Analytical validation and clinical utility of the Action OncoKitDx panel have been established [20]. Both the panel and the Data Genomics software 1.5.1 are CE-IVD certified.According to the manufacturer, the panel can detect alterations present at a minimum allele frequency of 5%. Detection may be compromised if the sequencing depth is below 200×. To achieve this detection threshold, a minimum tumor cellularity of 30% is recommended, along with input DNA quantity between 50 and 200 ngand a DNA Integrity Number (DIN) > 3. For MSI interpretation, the recommended tumor cellularity is also ≥30%. Samples with a lower cellularity or suboptimal DNA quality may still be analyzed, though with reduced sensitivity and specificity.

#### 2.3.3. Determination of Microsatellite Instability (MSI)

MSI status was assessed using two complementary techniques:NGS, as described above, using the Action OncoKitDx panel.Immunohistochemistry (IHC), by evaluating the expression of DNA mismatch repair (MMR) proteins MLH1, MSH2, MSH6, and PMS2.

#### 2.3.4. IHC Interpretation Criteria for MMR Protein Expression

Figure 1 shows a representative example of MMR protein expression in one of the analyzed tumors. The loss of nuclear expression of PMS2 and MLH1 (Figure 1A,B), along with preserved expression of MSH2 and MSH6 (Figure 1C,D), indicates a deficient MMR (dMMR) phenotype. This pattern suggests probable MLH1 inactivation, commonly due to promoter hypermethylation or gene mutation.

The simultaneous loss of MLH1 and PMS2 typically reflects MLH1 inactivation, since PMS2 stability and expression depend on MLH1—frequently observed in sporadic MSI tumors without *BRAF* mutation. In contrast, isolated PMS2 loss may indicate Lynch syndrome.

Figure 1A,B show an absence of nuclear staining for PMS2 and MLH1 in tumor cells, while Figure 1C,D display strong nuclear staining for MSH2 and MSH6, consistent with preserved expression.

Immunohistochemical evaluation of MMR proteins in sporadic CRC is a fundamental diagnostic tool. These findings underscore the clinical value of IHC in identifying MMR system deficiencies, with implications for prognosis and eligibility for targeted therapies.

Nuclear staining in tumor cells was interpreted as positive expression of the corresponding marker. Cases with preserved expression of all four proteins were classified as proficient MMR (pMMR), whereas a loss of expression of at least one protein was classified as deficient MMR (dMMR).

#### 2.3.5. Real-Time PCR: Idylla *KRAS* Mutation Test and Idylla *NRAS-BRAF* Mutation Test

*KRAS* mutation analysis was performed using the Idylla™ *KRAS* Mutation Test kit and the Idylla™ platform (Biocartis, Mechelen, Belgium). This is an in vitro diagnostic assay that enables qualitative detection of 21 mutations in codons 12, 13, 59, 61, 117, and 146 of the *KRAS* gene through a fully automated process that integrates target region amplification and real-time PCR-based detection.

The analysis of *NRAS* and *BRAF* mutations was conducted using the Idylla™ *NRAS-BRAF* Mutation Test kit and the Idylla™ platform (Biocartis). This in vitro diagnostic test allows for qualitative detection of 18 mutations in codons 12, 13, 59, 61, 117, and 146 of the *NRAS* gene, as well as 5 mutations in codon 600 of the *BRAF* gene, through an automated process involving target amplification and real-time PCR detection.

### 2.4. Statistical Analysis

A descriptive analysis was conducted to determine the frequency of mutations in *KRAS*, *NRAS*, and *BRAF*, as well as the presence of microsatellite instability (MSI). To assess associations between gene mutations and MSI status, the following statistical methods were used:Chi-square test (χ^2^) and Fisher’s exact test, depending on data distribution.Calculation of odds ratios (ORs) with 95% confidence intervals (95% CIs).A *p* < 0.05 was considered statistically significant.

All statistical analyses were performed using the Jamovi software version 2.6.24.0.

### 2.5. Artificial Inteligence

In order to optimize the presentation of the results, artificial intelligence tools (ChatGPT-4.5 and OpenAI-4.5) were employed to structure a table that clearly and accurately represented the findings from the descriptive analysis previously performed by the author. Additionally, these tools were used to support the interpretation and synthesis of the results obtained from the comparative analysis, once the statistical processing of the data had been completed.

## 3. Results

### 3.1. Descriptive Study

A descriptive analysis (*n* = 166; Figure 2) of the mutational frequency in sporadic colorectal cancer was conducted to explore the prognostic implications of *KRAS* mutations and the *BRAF* V600E mutation. Patients with hereditary non-polyposis colorectal cancer (Lynch syndrome) or with tumors located in the middle or lower rectum were excluded. The mutational frequency was analyzed in cases for which at least partial molecular data were available. A larger number of patients had individual MSI testing results compared to the other molecular markers, with *KRAS* testing being the next most frequent.

A descriptive statistical analysis was performed on 166 colorectal cancer (CRC) tumors. A high prevalence of *KRAS* mutations was observed (52.4%), highlighting its major role in colorectal tumorigenesis and its relevance to resistance to anti-EGFR therapies. The *NRAS* gene showed a low mutation rate (8.9%), consistent with its lower prevalence reported in the literature compared to *KRAS*. *BRAF* mutations were present in 22.1% of cases, a slightly higher proportion than typically described, potentially indicating a subgroup of patients with distinct clinical features, such as a greater association with MSI. Microsatellite instability was detected in 12.1% of tumors, a frequency consistent with expectations for sporadic colorectal cancer.

### 3.2. Comparative Study

A comparative analysis was conducted on those patients who met all the inclusion criteria, evaluating the association between each gene and the presence or absence of microsatellite instability.

#### 3.2.1. *KRAS* and MSI Association

First, the association between *KRAS* mutation and MSI status was assessed, yielding the following findings (Figure 3).

The statistical analysis performed (Table 1) revealed a significant association between *KRAS* mutations and microsatellite stability, with a *p* = 0.049 in the chi-square test. This suggests that tumors harboring *KRAS* mutations are less likely to exhibit microsatellite instability (MSI). However, when applying Fisher’s exact test—more appropriate in contingency tables with small sample sizes—the *p* = 0.070. Although close to the conventional threshold of significance, this result does not allow for a definitive conclusion, and the observed association may be due to chance (Table 2). Regarding the odds ratio analysis, an OR of 0.106 (95% CI: 0.00549–2.06) was observed, suggesting a lower probability of MSI in *KRAS*-mutated tumors. Nevertheless, the wide confidence interval crossing the null value indicates substantial uncertainty and a non-conclusive association (Table 3).

#### 3.2.2. *NRAS* and MSI Association

To evaluate the potential association between *NRAS* mutations and mismatch repair (MMR) deficiency, a similar statistical analysis was conducted, yielding the following results (Figure 4).

The chi-square test results for the association between *NRAS* mutation status and microsatellite stability (Table 4) showed no statistical significance, with a *p* = 0.509, far above the standard significance threshold (*p* < 0.05), precluding rejection of the null hypothesis. This implies no association between *NRAS* mutation and microsatellite status. Supporting this finding, Fisher’s exact test yielded a *p* = 1.000, further reinforcing the absence of a statistically significant relationship (Table 5). Moreover, the odds ratio was 0.896 (95% CI: 0.0405–19.8), a value very close to 1, indicating no appreciable association. The wide confidence interval that includes the null value again highlights the lack of statistical power and potential for random variation (Table 6).

#### 3.2.3. *BRAF* and MSI Association

Subsequently, the association between *BRAF* mutation (specifically the V600E variant) and MSI status was assessed, and the following results were obtained (Table 7 and Figure 5).

The chi-square test revealed a statistically significant association between *BRAF* mutations and microsatellite instability (MSI) (*p* = 0.043), indicating that tumors harboring *BRAF* mutations are more likely to exhibit MSI. This raises the possibility of *BRAF* serving as a biomarker for identifying MSI-positive tumors. However, Fisher’s exact test yielded a *p* = 0.078, which—although below 0.1—does not meet the conventional threshold for statistical significance (*p* < 0.05) (Table 8).

The odds ratio (OR) was 6.43 (95% CI: 0.897–46.1), which may suggest a considerable association between *BRAF* mutations and MSI. Nevertheless, the wide confidence interval, which includes 1, reflects a high degree of uncertainty and limits the robustness of this conclusion. Despite this, the elevated OR points to a potentially relevant trend that warrants further investigation in larger cohorts (Table 9).

## 4. Discussion

Our findings support the role of *KRAS* as the most prevalent mutation in sporadic colorectal cancer (sCRC), particularly associated with microsatellite-stable (MSS) tumors, while *BRAF*-mutated tumors show a higher frequency of MSI, consistent with their recognized clinical relevance. The low frequency of *NRAS* mutations aligns with previous studies and does not appear to be associated with MSI.

The most relevant findings include a high prevalence of *KRAS* mutations (52.4%), a substantial proportion of *BRAF* V600E mutations (20.8%), and a low *NRAS* mutation rate (8.9%), consistent with previous studies in the literature. MSI was identified in 12.1% of cases, aligning with expected values in the context of sporadic colorectal cancer.

From a statistical perspective, a significant association was observed between *KRAS* mutations and MSS status (*p* = 0.049), suggesting that *KRAS*-mutated tumors tend to have a functional mismatch repair (MMR) system. This finding is consistent with previous studies linking *KRAS* mutations with MSS tumors and the distinct molecular profiles of MSI phenotypes. Conversely, a significant association was identified between *BRAF* V600E mutations and MSI (*p* = 0.043), reinforcing *BRAF*’s role as a molecular marker of this tumor subtype. However, the small sample size and wide confidence intervals—such as the OR of 6.43 (95% CI: 0.897–46.1)—require cautious interpretation and further validation in larger studies. No significant association was found between *NRAS* mutations and MSI status (*p* = 0.509). The OR close to 1 (0.896) and the wide confidence interval crossing the null value suggest that *NRAS* mutations likely do not play a relevant role in the genomic instability of these tumors.

These findings are consistent with the results reported by Ye et al. (2015), described a higher prevalence of *KRAS* mutations in MSS tumors, supporting the hypothesis that *KRAS* mutations are associated with a functional MMR system [13]. Additionally, the low frequency of *NRAS* mutations in our cohort is in line with the same study, highlighting a more limited role for this gene in the tumorigenesis of sporadic colorectal cancer. The association between *BRAF* V600E mutations and the MSI phenotype has been widely reported in the literature. Specifically, studies by Birgisson et al. (2015) and Fan et al. (2021) have demonstrated a strong association between *BRAF* mutations and MSI-positive tumors, in agreement with our findings [4,5]. Moreover, Rasuck et al. (2012) emphasized the role of *BRAF* V600E in the context of genomic instability and MMR epigenetic alterations, underlining its value as a molecular and prognostic biomarker [21].

Despite the clinical and biological relevance of these findings, several limitations must be acknowledged. First, the small sample size—particularly in the subgroup with complete molecular and MSI data (*n* = 42)—limits the statistical power of the analyses and affects the robustness of the detected associations. This is reflected in the Fisher’s exact test results and the wide confidence intervals observed, indicating substantial variability in the estimates.

Additionally, the retrospective, single-center design may introduce selection bias. The lack of comprehensive clinical data in some cases—such as tumor stage, specific anatomical location, or treatment details—precluded a more detailed patient characterization and hindered the analysis of potential correlations between molecular profile and clinical behavior.

According to our decision model for gene testing, next-generation sequencing (NGS) proved to be superior to single-gene analysis techniques (real-time PCR and immunohistochemistry) [22]. Including additional relevant biomarkers—such as PIK3CA, NTRK alterations, or MMR-related epigenetic changes—could further refine the molecular profiling of tumors and enhance clinical decision-making in the 42 patients studied (Figure 6).

Therefore, future studies should aim to expand the patient cohort through a prospective, multicenter design that integrates clinical, molecular, and outcome variables. Validation in independent populations would enhance the generalizability and applicability of these findings, laying the groundwork for a more personalized approach in the management of sporadic colorectal cancer.

High-frequency microsatellite instability (MSI-H), associated with deficient mismatch repair (dMMR), is gaining increasing relevance as a biomarker in patients with advanced cancer to assess their eligibility for immune checkpoint inhibitor (ICI) therapies. Currently, multiple next-generation sequencing (NGS)-based methods are available to analyze MSI status. Comprehensive genomic profiling (CGP) tests enable the accurate determination of both MSI status and other genomic alterations through a single NGS-based assay, including liquid biopsy samples. As a result, MSI-H has been identified across various tumor types, promoting the wider use of immunotherapy, which appears effective in neoplasms characterized by a high mutational burden and/or neoantigens. Studies utilizing NGS have also helped characterize how MSI drives carcinogenesis, including a high incidence of kinase fusions in MSI-H colorectal cancer (CRC), which represent candidates for targeted kinase inhibitors [23]. NTRK gene fusions have been linked to the serrated pathway of colorectal cancer development. Oncogenic alterations in HER2 and its dimerization partner HER3 may influence responses to HER2-targeted therapies. Therefore, assessing these markers holds significant clinical value in the molecular profiling of colorectal cancer, as it allows identification of patient subgroups with potential therapeutic benefits. Recent advances in MSI-H tumor research have driven the development of new diagnostic tools and treatments, such as synthetic lethality strategies targeting the Werner gene. The detection of tumor DNA is essential to trigger antitumor immune responses in the presence of dMMR, opening up new opportunities to identify useful biomarkers for immunotherapy. In this context, the analysis of genomic alterations associated with MSI and MSI-H carries significant clinical relevance [24,25].

## 5. Conclusions

Based on these data, we conclude that initial molecular analysis using comprehensive NGS panels is more efficient, saving time, tissue, and costs compared to stepwise single-gene testing. Moreover, this approach allows for more accurate molecular classification and facilitates the selection of targeted therapies and immunotherapy. In the absence of upfront NGS availability, we suggest starting with *KRAS* analysis, followed by *BRAF* and MSI testing in *KRAS* wild-type cases, thus promoting an efficient molecular classification strategy for CRC.

## Figures and Tables

**Figure 1 cancers-17-02071-f001:**
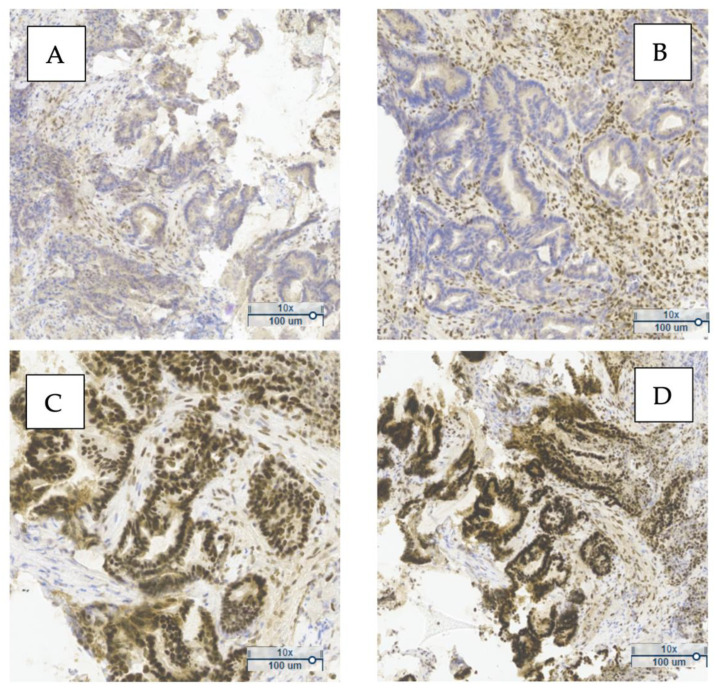
Immunohistochemical expression of mismatch repair (MMR) proteins in tumor tissue from patients with sporadic colorectal cancer. (**A**) Loss of nuclear expression of PMS2 with preserved staining in adjacent normal cells. (**B**) Loss of nuclear expression of MLH1 with positive staining in non-tumor cells (internal control). (**C**) Positive nuclear expression of MSH2. (**D**) Positive nuclear expression of MSH6. Brown-yellow staining indicates immunohistochemical positivity. Observations were performed using optical microscopy at 10× magnification.

**Figure 2 cancers-17-02071-f002:**
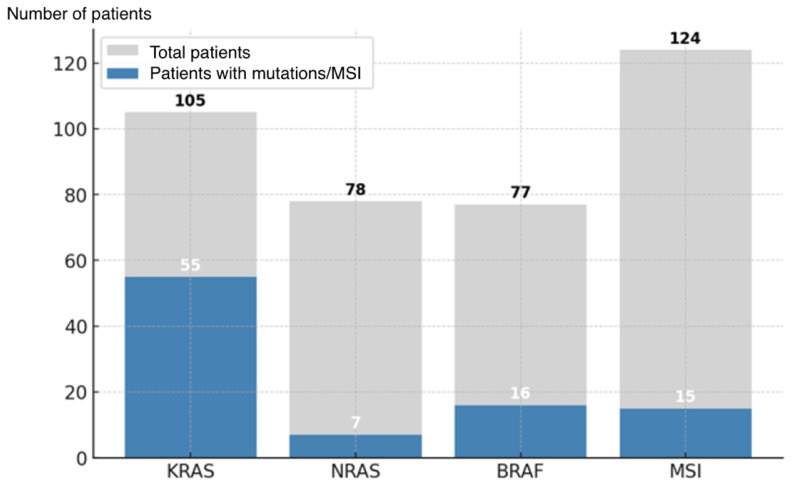
Prevalence of *KRAS*, *NRAS*, and *BRAF* mutations and microsatellite instability (MSI).

**Figure 3 cancers-17-02071-f003:**
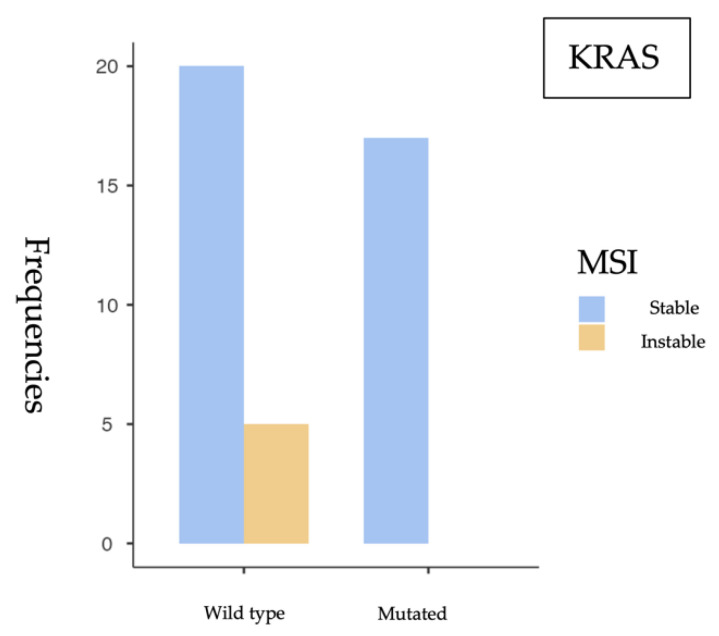
Frequency of *KRAS* mutations according to microsatellite instability status. No MSI cases were observed among *KRAS*-mutated tumors.

**Figure 4 cancers-17-02071-f004:**
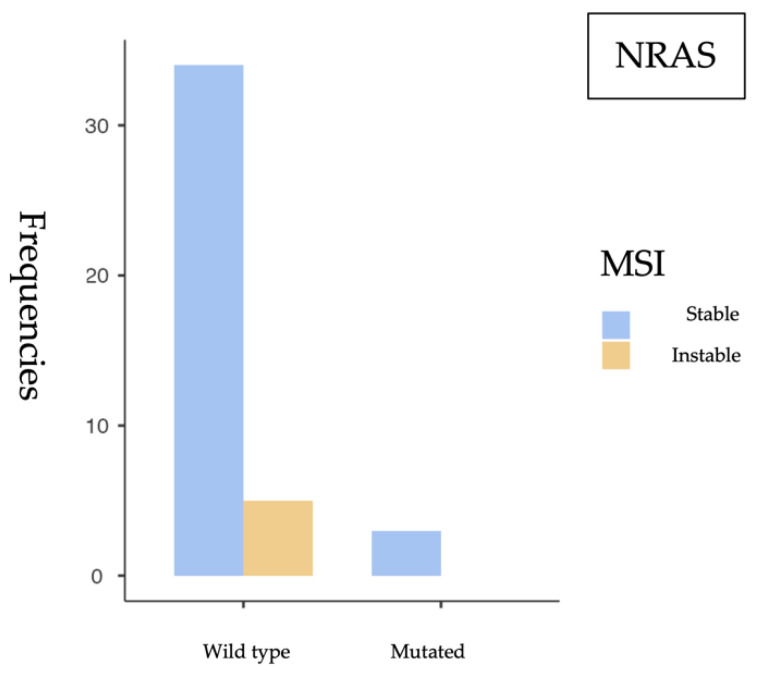
Frequency of *NRAS* mutations according to microsatellite instability status. No MSI cases were observed among *NRAS*-mutated tumors.

**Figure 5 cancers-17-02071-f005:**
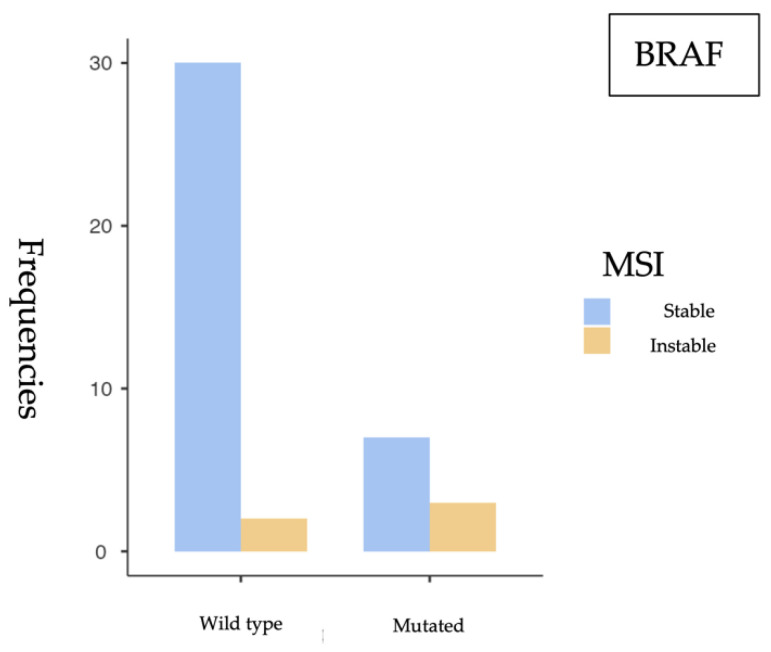
Frequency of *BRAF* mutations according to microsatellite instability status. MSI cases were observed among *BRAF*-mutated tumors.

**Figure 6 cancers-17-02071-f006:**
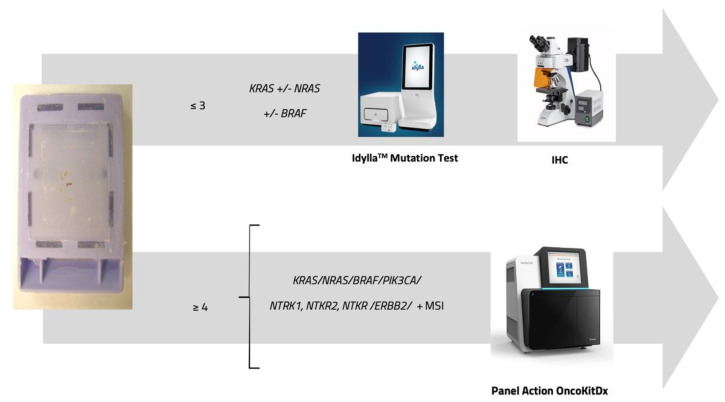
Overview of molecular alteration detection strategies comparing the two most commonly used diagnostic approaches. Top row: analysis of ≤3 genes via Idylla™ Mutation Test and subsequent MMR system evaluation via immunohistochemistry. Bottom row: comprehensive analysis of ≥4 genes and MMR status using NGS.

**Table 1 cancers-17-02071-t001:** Contingency table.

	MSI		
*KRAS*	Stable	Instable	Total
Wild-Type	20	5	25
Mutated	17	0	17
Total	37	5	42

**Table 2 cancers-17-02071-t002:** Chi-square test results.

	Value	Gl	*p*
X^2^	3.86	1	0.049
Fisher’s Exact Test *p*-value			0.070
N	42		

**Table 3 cancers-17-02071-t003:** Comparative measures.

		95% Confidence Intervals	
	Value	Lower	Upper
Odds Ratio	0.106 ^1^	0.00549	2.06
Relative Risk	0.800 ^2^	0.658	0.973

^1^ Haldane–Anscombe correction applied. ^2^ Row comparisons.

**Table 4 cancers-17-02071-t004:** Contingency table.

	MSI		
*NRAS*	Stable	Instable	Total
Wild-Type	34	5	39
Mutated	3	0	3
Total	37	5	42

**Table 5 cancers-17-02071-t005:** Chi-square test results.

	Value	Gl	*p*
X^2^	0.437	1	0.509
Fisher’s Exact Test *p*-value			1.000
N	42		

**Table 6 cancers-17-02071-t006:** Comparative measures.

		95% Confidence Intervals	
	Value	Lower	Upper
Odds Ratio	0.896 ^1^	0.0405	19.8
Relative Risk	0.872 ^2^	0.773	0.983

^1^ Haldane–Anscombe correction applied. ^2^ Row comparisons.

**Table 7 cancers-17-02071-t007:** Contingency table.

	MSI		
*BRAF*	Stable	Instable	Total
Wild-Type	30	2	32
Mutated	7	3	10
Total	37	5	42

**Table 8 cancers-17-02071-t008:** Chi-square test results.

	Value	Gl	*p*
X^2^	4.10	1	0.043
Fisher’s Exact Test *p*-value			0.078
N	42		

**Table 9 cancers-17-02071-t009:** Comparative measures.

		95% Confidence Intervals	
	Value	Lower	Upper
Odds Ratio	6.43	0.897	46.1
Relative Risk	1.34 ^1^	0.884	2.03

^1^ Row comparisons.

## Data Availability

Data sharing is not applicable, due no new data were created or analyzed in this study. Data sharing is not applicable to this article.

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
