# Peer review of "Elucidating the Role of KRAS, NRAS, and BRAF Mutations and Microsatellite Instability in Colorectal Cancer via Next-Generation Sequencing"

_cancers, 2025, doi:10.3390/cancers17132071_

Round 1

Reviewer 1 Report

Comments and Suggestions for Authors

The manuscript presents relevant findings regarding the molecular characterization of sporadic colorectal cancer, particularly the associations between key oncogenic mutations and microsatellite status. The study aligns well with existing literature and contributes to the understanding of molecular diagnostics. However, improving the clarity of presentation, enhancing the visual quality and consistency of figures, and clearly defining the study’s unique contributions will significantly strengthen its impact and readability.

(Title):
The current title is excessively lengthy and a little repetitious.

(Abstract):

  • "Comprehensive NGS profiling from the start" is an important part; try emphasizing this early or more strongly.

(Keywords):

  • Please write the "keywords" based on alphabetically.
  • Change "mismatch repair system" to "DNA mismatch repair" to conform to conventional language.

(Figure 1):

  • While magnification (10×) is noted, consider including a scale bar in the figure itself.
  • Please ensure that all terms throughout the manuscript including those in figure legends and figure labels are written in English. There are instances where words such as “mutado” appear, which are not in English.

(Line 254):

  • Ensure consistent formatting of statistical values: p = 0.049 (not “p-value of 0.049”).

(Line 307):

  • The term "suggests a strong association" between BRAF and MSI based on a non-significant Fisher's test and a large OR CI crossing 1 could be misleading. This should be adjusted to reflect the exploratory character of the findings and the need for validation in larger samples.

Major Concern about Discussion and Conclusion:

  • The statistical correlations between KRAS and MSS (p = 0.049) and BRAF and MSI (p = 0.043) are barely significant and based on small subgroups with large confidence intervals. The findings should be interpreted with caution, particularly when drawing comparisons to known literature.
  • While the findings are consistent with previous research (Ye et al., Birgisson et al., Fan et al.), it is unclear what new addition this study makes other than validating existing knowledge. The authors should clarify the unique value of their data.
  • Figure 6 is mentioned in the discussion but not adequately explained. Also, quality of this figure needs to improve.

Minor Comments:

  • While the discussion emphasizes the lack of a correlation between NRAS and MSI, a brief comment on its recognized biological role in sCRC development would help to complete the interpretation.

Reviewer 2 Report

Comments and Suggestions for Authors

MANUSCRIPT: 3681934

TITLE: Molecular Characterization of Sporadic Colorectal Cancer: Association Between KRAS, NRAS, and BRAF Mutations and Microsatellite Instability via Next-Generation Sequencing

The manuscript 3681934 “Molecular Characterization of Sporadic Colorectal Cancer: Association Between KRAS, NRAS, and BRAF Mutations and Microsatellite Instability via Next-Generation Sequencing”, presents an interesting study in order to use a more recent method, Next-Generation Sequencing (NGS), in comparison with other methods used such as polymerase chain reaction (PCR) or immunohistochemistry (IHC) in order to evaluate or classify the association Between KRAS, NRAS, and BRAF Mutations and Microsatellite.

The presented work is well structured, well planned and the research is competently carried out.

Methodology used was adequate to the research objectives

Literature cited is adequate and more than 50% are from the last five years, however the number of references used are scarce.

Statistical analysis was performed and adequate to the work.

Results and discussion are properly discussed and conclusions are presented according to the results obtained.

However, some questions remain to be clarified and solved and the manuscript in its current form must be revised in minor several points as follows comments:

  1. Please recommend increasing the number of references since the manuscript presents only 16 references which is a scarce number.
  2. Please review affiliations 1 and 2, an affiliation is not the email address as presented in the manuscript.

Reviewer 3 Report

Comments and Suggestions for Authors

This review and re-analysis of previous data includes a significant cohort, and the analyses performed are well-designed and support the proposed conclusions. The study's limitations are well-stated and relevant, considering that it used a somewhat limited set of biomarkers.
Regarding the text, it is worth reviewing as some abbreviations appear before their explicitation, and some references are unformatted (they include the DOI).
There are also some typographical errors, likely the result of the review.
